# Extreme Climate Shocks and Green Agricultural Development: Evidence from the 2008 Snow Disaster in China

**DOI:** 10.3390/ijerph182212055

**Published:** 2021-11-17

**Authors:** Litao Feng, Zhuo Li, Zhihui Zhao

**Affiliations:** School of Economics and Management, Wuhan University, Wuhan 430072, China; lmx19950213@whu.edu.cn

**Keywords:** extreme climate shocks, agricultural green development, environmental pollution, entropy weight method

## Abstract

Extreme climate shocks cause agricultural yield reductions and increase long-term climate risk, altering farmers’ long-term production decisions and affecting green agricultural development (GAD). We take the 2008 snow disaster in China as an extreme climate shock, calculate the GAD index by the entropy weighting method, and use the difference-in-difference method to study the extreme climate shock’s impact on GAD. The results show that: (1) Extreme climate shocks are detrimental to GAD, with the snow disaster decreasing China’s GAD level by 3.07%. (2) The impacts of extreme climate shocks are heterogeneous across climate and economic zones, with greater impact in humid and developed regions. (3) Extreme climate shocks affect GAD mainly by reducing farmers’ willingness to cultivate, and increasing energy consumption, fertilizer, and pesticide input. (4) Extreme climate shocks do not reduce agricultural yields in the long run. Still, they reduce the total value of agricultural production and decrease the quality of agricultural products expressed in terms of unit value. The findings of this study have policy implications for developing countries in coping with extreme climate shocks and promoting GAD.

## 1. Introduction

With the increase in global warming and pollution, green development has become a common goal pursued by many countries. Agriculture is constrained as a sector closely related to the natural environment, while its production methods can significantly affect the environment and climate. Pursuing GAD and reduced land and water resources by fertilizers and pesticides are important issues for developing countries, including China.

An important way to achieve a green revolution in agriculture is to increase the long-term benefits of green agriculture for farmers [1]. In the long term, uncertainty from climate change is an important constraint to agricultural investment and green development. Extreme climate shocks will bring huge exogenous shocks to agricultural production and lead to long-term climate risks in the affected areas. This impact can change farmers’ production methods, because rising climate risks will reduce long-term expected returns, leading farmers to prefer short-term production and reducing their willingness to engage in green production. The study of the impact of extreme climate shocks on GAD is of great theoretical and practical significance for realizing the green revolution in developing countries. However, most scholars have focused on the factors influencing GAD [2,3] or the impact of climate change and disasters on agricultural production [4]. There are relatively few studies examining the impact of extreme climate shocks on GAD.

The 2008 snow disaster in China is one of the major meteorological disasters suffered by China in the 21st century. The abnormal atmospheric circulation caused by the La Niña phenomenon in 2007 led to intense rainfall in southern China and triggered snow disasters in early 2008 [5,6]. The snowstorm affected 20 provinces in the north and south of China, with 11.87 million hectares of crops affected and 1.69 million hectares of crop failure. This disaster is one of China’s most serious climate disasters in the 21st century and has greatly hindered China’s agricultural development. The snowstorm reduced farmers’ incomes, destroyed much of the agricultural infrastructure, and forced many farmers to work outside. As farmers did not expect such a serious climate disaster in winter, this snow disaster has increased the future climate risk, increased the cost and risk of green agriculture, and affected farmers’ future production decisions. The 2008 snow disaster in China provides a natural experiment for us to study the relationship between extreme climate shocks and GAD.

We conduct an empirical analysis using a sample of 2086 counties in China in the period of 2000–2018 to study the impact of extreme climate shocks on GAD. To measure the level of GAD, we refer to the existing literature [7,8,9] and use the entropy weight method (EWM) to calculate the composite indicators of GAD. The empirical analysis was conducted using a difference-in-difference (DID) method based on the degree of damage to each county during the 2008 snow disaster. On this basis, heterogeneity analysis was conducted, and the mechanism was analyzed in terms of willingness to cultivate, production input, and agricultural pollution.

This paper has the following innovations: (1) We examined the impact factors of GAD from an extreme climate perspective and provided empirical evidence that extreme climate shocks impede GAD in the long term. (2) We found heterogeneity in climate extremes’ impact on GAD due to climate and economic development differences. In humid zones and economically developed areas, extreme climates have a greater impact on GAD. (3) We investigated the impact mechanism of extreme climate shocks on GAD and found that extreme climate shocks mainly hinder GAD by reducing farmers’ willingness to cultivate, increasing energy consumption in agricultural production, and increasing fertilizer and pesticide inputs. At the same time, we found that farmers increase fertilizer and pesticide inputs to make up for short-term losses, which protects yields but reduces the proportion of green products in the long run and reduces the quality of agricultural products.

## 2. Literature Review and Theoretical Analysis

### 2.1. Literature Review

There is little literature on the impact of extreme climate shocks on GAD. The relevant studies mainly focus on three areas: GAD, climate change and green development, and the impact of climate change on agricultural production.

In terms of GAD, scholars believe that the connotation of GAD is the combination of agricultural productivity improvement and environmental protection [2,10,11,12]. As increasing agricultural income is a prerequisite for green agricultural production, investment in land and intensification are important [3]. Fertilizers and pesticides are the most important input factors in the influence of GAD [13]. Recent studies have concluded that soil N_2_O emissions are also an important indicator for measuring the level of GAD [14]. Some scholars believe that the impact of transgenic technology on GAD is uncertain [15]. In contrast, others believe that new plant breeding technologies (NPBTs) are an important way to promote GAD [16]. In terms of the green revolution in agriculture, many scholars believe that the standards of the green revolution in agriculture at this stage cannot be implemented in developing countries [17,18].

In the area of climate change and green development, scholars believe that climate change exacerbates the risk of agricultural production, thus it is harmful to water resources [19], hygiene [20], and ecology [19,21]. In terms of coping strategies, existing research considers that green adaptation strategies can make cities more tolerant of climate change stresses [22], and climate-smart agriculture is beneficial to GAD [23]. Meanwhile, the issuance of green bonds is also an effective strategy [24]. However, the current allocation of funds for international organizations to combat climate change does not sufficiently consider all parties’ needs [25]. At the same time, some scholars believe that social and political barriers are important reasons for the slow green development [26].

Current research remains diverse on the impact of climate change on agricultural production. Most scholars believe climate change will increase extreme weather [27,28] and reduce agricultural output [4]. Representative findings include that increased temperature and reduced rainfall due to climate change will reduce agricultural output in arid regions [29,30], and abnormal increases in rainfall in wet areas can also hinder crop growth [31,32]. Therefore, the negative impact of climate change on agriculture is significantly heterogeneous across regions [33]. However, some scholars disagree. Some scholars who have studied wheat cultivation in the United States believe that the effects of climate change are overestimated [34]. At the same time, other scholars believe that climate change is an important factor in promoting agricultural restructuring and expansion [35]. The divergence of scholars on the impact of climate change indicates the necessity of in-depth research on the impact of extreme climate shocks on GAD.

The existing literature is deficient in the following aspects. First, there is a lack of literature directly studying the impact of extreme climate shocks on GAD. Only a few scholars have studied the impact of extreme weather on agricultural efficiency from a sustainable development perspective [36,37]. However, the existing literature on the measurement and influencing factors of the level of GAD provides the foundation for measuring GAD. The studies on the impact of climate change provide the theoretical foundation. These results make it possible to study the impact of extreme climate shocks on the green development of agriculture. Second, the research methodology is insufficient, as the existing literature is more generalized and case based. At the same time, the relevant empirical analysis does not sufficiently consider the endogeneity issue, therefore, more rigorous empirical evidence is needed. Based on the above reasons, the 2008 snow disaster in China was taken as an extreme climate shock. The impact of extreme climate shock on GAD was studied by constructing a GAD index through the entropy weight method. In the empirical method, we use the difference-in-difference method (DID) to solve the endogeneity problem, obtain the causal effect of extreme climate shocks on GAD, and conduct robustness tests in several aspects to guarantee credibility of the results.

### 2.2. Theoretical Analysis: How Extreme Climate Shocks Affect GAD

Extreme climate shocks reduce agricultural output in the short term and increase long-term climate risks [38]. Since agricultural production cycles are long and dependent on the climate, extreme climate shocks can significantly affect farmers’ production decisions and GAD [39]. Second, the rise of long-term risks in agriculture makes farmers who continue to engage in agriculture more inclined to secure short-term output [40], which leads to increased energy consumption and pollution in agriculture, ultimately reducing the level of GAD (shown in Appendix A Figure A1). Figure 1 shows the mechanisms of extreme climate shocks on GAD.

#### 2.2.1. Decreasing Willingness to Cultivate

Extreme weather shocks first affect farmers’ willingness to cultivate. Unlike climate change, which affects agricultural production in the long term, extreme climate shocks can wreak havoc on agricultural production in the short term by reducing agricultural yields and damaging some agricultural facilities [41] and reducing agricultural incomes and production conditions [40]. Extreme climate shocks increase the GAD between agricultural and non-agricultural incomes, leading some farmers to choose to work outside and transfer or abandon their land [38], thus reducing the utilization rate of arable land. The lack of maintenance of abandoned land leads to soil erosion, which continues to discourage GAD.

#### 2.2.2. Increasing Agricultural Energy Consumption

For agricultural production, water is the most important input resource. Due to the scarcity of water resources, developing water-saving agriculture in areas lacking large amounts of rainfall is conducive to protecting groundwater resources and the environment, preventing land desertification. It is one of the most important measures for GAD. Water-saving agriculture is conducive to sustainable development in the long term but can reduce yields in the short term. After facing extreme weather shocks, farmers may abandon water-saving agriculture to compensate for losses and turn to large amounts of river and groundwater extraction for irrigation [42], resulting in increased energy consumption for agricultural production [36]. The pumps required for pumping water rely on electricity and diesel drive. Their massive use is detrimental to water conservation and increases environmental pollution [43], thus reducing GAD.

#### 2.2.3. Increasing Agricultural Pollution

Reducing fertilizer and pesticide inputs to protect the environment is one of the core connotations of GAD. Although reducing fertilizer and pesticide inputs will reduce agricultural yields in the short term, in the long term, it will help safeguard soil fertility and provide green agricultural products, thus improving product quality and prices and ultimately increasing farmers’ income. An important factor in the promotion of green agriculture is the increase in long-term benefits [1]. When extreme climate shocks occur, the long-term future climate risk increases and the expected rate of return of green agriculture decreases [40], thus reducing farmers’ willingness to develop green agriculture. When farmers face stronger future climate risks, they focus more on short-term gains [39], which will increase fertilizer and pesticide inputs [11], exacerbate agricultural pollution, and reduce the level of GAD.

## 3. Methods

### 3.1. Difference-in-Difference Model

We use the difference-in-difference method to analyze the impact of the snow disaster on the level of GAD, and the regression equation is as follows:(1)$$green_{it}=\beta_{0}+\beta_{1}Rain_{i}^{s}\times Post_{t}+X_{it}+\delta_{i}+\gamma_{t}+\varepsilon_{it}$$
where $green_{it}$ is the dependent variable, representing the GAD index of county i in year t. Referring to the classic literature [44], we use rainfall fluctuations $Rain_{i}^{s}$ in each county during the snowstorm (January 2008) as a proxy variable for the degree of disaster. $Post_{t}$ is the time treatment variable and, when the time is before 2008, $Post_{t}$ equals 0, and vice versa, it equals 1; $X_{it}$ is the control variable; $\delta_{i}$ and $\gamma_{t}$ represent county fixed effects and time fixed effects; $\varepsilon_{it}$ represents the error term.

### 3.2. Calculation of GAD Index

We use the entropy weighting method (EWM) to calculate a comprehensive index of the level of GAD. Combining previous scholars’ studies [45,46] and agricultural production backgrounds, we calculate the GAD index from five aspects: Agricultural endowment, agricultural production efficiency, agricultural production energy consumption, agricultural pollution, and environmental protection. The details and definitions of all variables are shown in Table 1.

Referring to previous scholars [47], we calculate the GAD index in four steps.

The first step is to standardize all metrics, where j represents the jth variable.
(2)$$\mathrm{Positive}\;\mathrm{indicators}:\;z_{itj}=\frac{y_{itj}-min_{j}\left\{y_{itj}\right\}}{max_{j}\left\{y_{itj}\right\}-min_{j}\left\{y_{itj}\right\}}$$
(3)$$\mathrm{Negative}\;\mathrm{indicators}:\;z_{itj}=\frac{max_{j}\left\{y_{itj}\right\}-y_{itj}}{max_{j}\left\{y_{itj}\right\}-min_{j}\left\{y_{itj}\right\}}$$

The second step is to calculate the entropy value.
(4)$$e_{it}=-k\sum_{j=1}^{n}p_{itj}lnp_{itj}$$
(5)$$p_{itj}=\frac{z_{itj}}{\sum_{j=1}^{n}z_{itj}},k=\frac{1}{lnn}$$

The third step is to calculate the entropy weights.
(6)$$w_{it}=\frac{1-e_{it}}{T-\sum_{t=1}^{T}e_{it}}\left(0\leq w_{it}\leq 1,\sum_{t=1}^{T}w_{it}=1\right)$$

The fourth step is to calculate the comprehensive index of GAD.
(7)$$green_{it}=\sum_{j=1}^{n}w_{it}z_{itj}$$

### 3.3. Independent Variables

We take the DID variable $Rain_{i}^{s}\times Post_{t}$ as the core independent variable. The coefficient $\beta_{1}$ represents the effect of the snow disaster on the dependent variable $green_{it}$. Different from the standard DID, this paper uses a continuous variable: Rainfall fluctuations $Rain_{i}^{s}$ of each county during the snow disaster (January 2008) as a variable to identify its degree of disaster, thus better identifying the impact of the snow disaster on each region. A larger $\;Rain_{i}^{s}$ represents more rainfall in the county during January 2008. Since this is the coldest month of the year, an abnormal increase in rainfall would increase snowfall in the area, thus exacerbating its risk of snow disaster. We use the z-score method to calculate the rainfall fluctuations of each county during the snow disaster.
(8)$$Rain_{i}^{s}=\frac{rain_{i}^{s}-{\bar{rain}}_{i}^{s}}{\sigma_{i}^{s}}$$
where $rain_{i}^{s}$ represents the total rainfall of county i in January 2008, ${\bar{rain}}_{i}^{s}$ and $\sigma_{i}^{s}$ represent the mean and standard deviation of rainfall of county i in January from 2000 to 2007.

### 3.4. Control Variables

Although the 2008 snow disaster was a manifestation of climate change, the degree of disaster in different regions may still be related to their economic and social characteristics. Developed regions may have better responses to reduce losses. Therefore, we consider factors that are potentially relevant to both disaster and GAD as control variables: Total population (POP), agricultural machinery power (Machinery), primary industry value-added (First), secondary industry value-added (Second), total government expenditure (Expand), and total social investment (Invest). All control variables are taken as logarithms in the empirical analysis. County-level fixed effects control factors that do not change with time during the sample period, such as terrain and slope. Time fixed effects control factors that do not change with county-level factors such as sea level rise and the world economic crisis.

## 4. Data Sources and Descriptive Statistics

### 4.1. Sample and Data Source

We use a sample of 2086 counties in China from 2000 to 2018 for empirical analysis and use the entropy weighting method (EWM) to construct the variables in Table 1 into a GAD index. Each variable used to construct the GAD index is obtained from the China County Statistical Yearbook, the China Statistical Yearbook, the China Rural Statistical Yearbook, and the Statistical Yearbook of every province. The China County Statistical Yearbook contains data on control variables such as population, agricultural machinery power, primary industry output value, secondary industry output value, government expenditure, and total social investment.

Rainfall data are from the FLDAS Noah Land Surface Model L4 Global Monthly 0.1 × 0.1 degree (FLDAS). We first obtain the monthly rainfall for each county from 2000–2018 based on latitude and longitude matching and then use the z-score to calculate the annual rainfall fluctuations for each county.
(9)$$Rain_{it}^{total}=\overset{12}{\underset{m=1}{\sum }}\frac{rain_{mit}-{\bar{rain}}_{mi}}{\sigma_{mi}}$$
(10)$$Rain_{it}^{wet}=\underset{m=wet}{\sum }\frac{rain_{mit}-{\bar{rain}}_{mi}}{\sigma_{mi}}$$
where i represents the county, t represents the year, and m represents the month. $Rain_{it}^{total}$ represents the total rainfall fluctuations of county i in year t. $Rain_{it}^{wet}$ represents the total rainfall fluctuations in the rainy season of county i in year t. $rain_{mit}$ represents the total rainfall in month m of county i in year t. ${\bar{rain}}_{mi}$ and $\sigma_{mi}$ represent the mean and standard deviation of rainfall of county i in month m from 2000–2018. The main variable definitions and data sources are shown in Table 2.

### 4.2. Descriptive Statistics

Since we identified the degree of damage in each county based on the rainfall fluctuations during the snow disaster (January 2008), to better demonstrate the balance of the sample, we treated the counties whose rainfall fluctuates above the median (0.3) during the snow disaster as the treatment group and the rest of the counties as the control group. Descriptive statistics are shown in Table 3. We can see from Table 3 that the difference between the main variables of the treatment group and the control group is relatively small, which indicates that whether each county or city is affected by the snow disaster has nothing to do with its own economic characteristics. The samples of the control group and the control group have good stability, which can ensure the validity of the empirical results.

## 5. Results

### 5.1. Baseline Estimations

The baseline estimations are shown in Table 4. Columns (1)–(5) take the interaction terms of the January 2008 rainfall fluctuations $Rain_{i}^{s}$ and the time dummy variable $Post_{t}$ as the DID variables, and report the estimations including different fixed effects and control variables. At the same time, to verify the credibility of the results, we consider the counties with rainfall fluctuations greater than 0.3 in January 2008 as the treatment group, and the remaining counties as the control group. After interacting with the time dummy variable $Post_{t}$, a new DID variable is formed. The specific results are shown in Table 4, column (6). At the same time, the results of the control variables show that the coefficients of total population (POP), secondary industry value-added (Second), and total social investment (Invest) are not significant, indicating that these factors have no effect on the level of GAD. The coefficients of agricultural machinery power (Machinery), primary industry value-added (First), and total government expenditure (Expand) are significant. However, after controlling for these factors, the coefficient of the explanatory variable $Rain_{i}^{s}\times Post_{t}$ did not change significantly, indicating that these factors will not lead to biases in the baseline results.

Column (1) reports the results that do not contain the control variables and only control the time fixed effects. The results show that the coefficient of $Rain_{i}^{s}\times Post_{t}$ is −0.008 and is significant at the level of 1%, which preliminarily indicates that the snow disaster has reduced the level of GAD. Column (2) reports the results after adding county-level fixed effects. The results show that the coefficient of $Rain_{i}^{s}\times Post_{t}$ is −0.013 and is significant at the level of 1%, indicating that the negative impact of the snow disaster on the level of GAD is related to the characteristics of each county, which may be heterogeneous. Columns (3)–(5) are the results after adding different control variables. The results show that the magnitude and significance of the coefficient of $Rain_{i}^{s}\times Post_{t}$ change very little, indicating that the impact of the snow disaster is very exogenous. Column (6) uses different Rain*Post variables for analysis, and the results show that the coefficient is still significantly negative, which preliminarily shows that the baseline estimations are robust. However, considering that the counties and cities with rainfall fluctuations around 0.3 in column (6) are artificially divided into treatment groups and control groups, the impact of the snow disaster will be overestimated. Therefore, we mainly analyze column (5) in the baseline results.

The baseline regression results show that the snow disaster has significantly reduced the level of GAD. Taking the result of column (5) as an example, it shows that the snow disaster caused an average decrease of 0.012 in the GAD index in the long run. Based on the average value of the GAD index of 0.39 in the total sample, the level of GAD has dropped by 3.07%. At the same time, we found that other control variables have little effect on the level of green agriculture except for agriculture-related variables. This result shows that the GAD does not completely depend on the overall economic level and government intervention is relatively independent.

### 5.2. Robust Checks

To further verify the robustness of the baseline estimation, we conduct robust checks from the following ten aspects.

#### 5.2.1. Parallel Trend

To test the effectiveness of the DID method, we conducted a parallel trend test on the sample, and the results are shown in Figure 2.

The results in Figure 2 show that before the snow disaster, there was no significant difference in the level of GAD between the control group and the treatment group. After the snow disaster, the GAD level of the treatment group was significantly lower than that of the control group in the second and sixth phases. This result shows that there is no prior effect, which satisfies the parallel trend hypothesis.

#### 5.2.2. Adding Time Trend

Some scholars believe that the time trend item is of great significance in the time series [10], and adding the time trend item can better control some factors that change over time. Column (1) in Table 5 reports the results after adding the time trend item. The results show that at the 1% level, the coefficient of $Rain_{i}^{s}\times Post_{t}$ is significantly negative, and the coefficient value (−0.012) is the same as the baseline estimations.

#### 5.2.3. Excluding Municipalities

As China’s four municipalities (Beijing, Tianjin, Shanghai, and Chongqing) have been highly urbanized, the proportion of agriculture and the employed population are limited. To avoid the impact of sample differences on the results, we excluded the county samples belonging to these four municipalities. The results are shown in column (2) of Table 5. The results show that after excluding the samples of municipalities, the coefficient of $Rain_{i}^{s}\times Post_{t}$ is −0.012, which is significant at the level of 1%, and the impact of the snow disaster on the level of GAD is still significantly negative, which indicates that the impact is not due to sample differences.

#### 5.2.4. Concurrent Events

During the sample period (2000–2018), the Chinese government also implemented a land transfer policy that promoted land resource optimization. As the land transfer policy has improved the agricultural scale, it is conducive to long-term agricultural investment and green production. If the counties affected by the snow disaster also implemented the land transfer policy later, it would have a confounding effect. To eliminate this confounding effect, we constructed the land transfer policy variable based on when each province promulgated the Law on Land Contract as the cut-off point and added it to the regression. The estimated results are shown in Table 5, column (3). The results show that after adding the land circulation variable, the coefficient of $Rain_{i}^{s}\times Post_{t}$ is −0.011, which is significant at the 1% level. It is consistent with the baseline results and shows that the impact of the snow disaster on GAD does not come from the confounding effects of land transfer policies.

#### 5.2.5. Climatic Factors

Although the snow disaster is a purely exogenous climate shock, the extent of the disaster in each region is closely related to its rainfall fluctuations. Recent studies by some scholars have found that rainfall fluctuations are no longer a random process due to climate change but have a time trend [4]. The rainfall fluctuations significantly impact agricultural production [47,48], so long-term rainfall fluctuations may be related to the degree of disaster and the level of GAD. We calculated the annual total rainfall fluctuations by Equation (9) and rainy season rainfall fluctuations by Equation (10) in each county from 2000 to 2018, and used them as control variables to control this influence. The results are shown in Table 5, columns (4)–(5). The results show that after adding rainfall fluctuation variables, the coefficient of $Rain_{i}^{s}\times Post_{t}$ is −0.012, which is significant at the 1% level. This result is the same as the baseline results, indicating that this impact did not have a confounding effect due to rainfall fluctuations. We also considered the impacts of other factors such as effective irrigation rate (see Table A1) and education (see Table A2) on GAD, and the results are still robust.

#### 5.2.6. Propensity Score Matching Method (PSM)

The main areas affected by the 2008 snow disaster in China were southern provinces. These places are mostly hilly areas that are not conducive to large-scale production and are traditional family farming areas. Therefore, there may be sample self-selection, that is, the affected counties are also counties with a low degree of green agriculture. To avoid self-selection bias, we selected samples by the PSM. We used Machinery, First, Second, Expend, and Invest as matching variables and then matched the treatment group and the control group for 1:1 neighbor matching to obtain new samples. The results of the PSM are shown in column (1) of Table 6.

The results in column (1) of Table 6 show that the coefficient of $Rain_{i}^{s}\times Post_{t}$ is −0.013, which is significant at the 1% level. The impact of the snow disaster on the level of GAD is still significantly negative, which indicates that the baseline estimations are not due to the self-selection of the sample. The snow disaster has reduced the level of GAD because extreme climate shocks have changed farmers’ production behaviors.

#### 5.2.7. Replace Independent Variables

The baseline estimate is to identify the affected area based on the rainfall fluctuations of the affected month. This guarantees the exogeneity of the independent variables, but it cannot fully reflect the disaster situation in different regions. Some regions may have large rainfall fluctuations but are not affected. To test the validity of the baseline estimation, in this section, we change the DID identification method. Based on the actual disaster situation of the Chinese provinces in the 2008 snow disaster, we analyzed the seven most severely affected provinces (Anhui, Jiangxi, Hubei, Hunan, Guangxi, Sichuan, and Guizhou) as the treatment group and the others as the control group. The results are shown in column (2) of Table 6. The results show that after changing the DID identification method, the coefficient of $Rain_{i}^{s}\times Post_{t}$ is −0.024, and the impact of the snow disaster on the level of GAD is still significantly negative, which indicates that the baseline results are robust.

#### 5.2.8. Instrumental Variable Estimation

To test the exogeneity of DID identification, in this section, we use instrumental variable estimation for discussion. We use the 2007 rainfall fluctuations and the latitude and longitude as the instrumental variables (IVs) of the January 2008 rainfall fluctuations, and use the two-stage least squares method (2SLS) for analysis.

First of all, the feasibility of using the 2007 rainfall fluctuations as the IV is shown by the La Niña phenomenon that appeared in 2007 [5,6], which caused the precipitation in the north and south of China to be much lower than normal. At the same time, as an ex ante factor, there is no two-way causality between the La Niña phenomenon and the snow disaster. At the same time, the La Niña phenomenon occurs in the Pacific Ocean, which is far away from China, and is highly exogenous. Therefore, the rainfall fluctuations in 2007 can be regarded as an IV of the rainfall fluctuations in January 2008.

Secondly, the reason for adopting latitude and longitude as an IV is that the snow disaster in 2008 was mainly concentrated in central China and south China. The snow disaster is regional, so the disaster areas are significantly related to the geographical location. The latitude and longitude are a completely exogenous factor, which guarantees the exogeneity of the IV.

The results using 2007 rainfall fluctuations as an IV are shown in column (3) of Table 6. The coefficient of $Rain_{i}^{s}\times Post_{t}$ is −0.034, which is significant at the 1% level. The results using latitude and longitude as an IV are shown in column (4) of Table 6. The coefficient of $Rain_{i}^{s}\times Post_{t}$ is −0.041, which is significant at the 1% level. The results show that after using two IVs for 2SLS, the impact of the snow disaster on the level of GAD is still significantly negative. These results show that the conclusion that the snow disaster reduced the level of GAD is robust.

#### 5.2.9. Replace the Dependent Variable

In this part, we adopt another calculation method and construct a new GAD index as the dependent variable for analysis. Specifically, we used two variables of agricultural production consumption and agricultural pollution in Table 1 to construct a new GAD index. This calculation method can eliminate the impact of agricultural productivity and simply measure the degree of agricultural production consumption and pollution. The specific results are shown in Table 6, column (5).

After adopting the new GAD index, the coefficient of $Rain_{i}^{s}\times Post_{t}$ is −0.006 and is significant at the 1% level. This result shows that after excluding factors such as agricultural productivity and agricultural endowment, the snow disaster has increased agricultural energy consumption and pollution, thereby reducing the level of GAD. It also shows that the negative impact of the snow disaster on the level of GAD is robust.

#### 5.2.10. Placebo Tests

To test whether the baseline results were caused by some missing random variables, we randomly divided the sample counties into a treatment group and a control group for placebo tests. The sample includes 2086 counties from 31 provinces in China, of which seven provinces were the most affected in the 2008 snow disaster. We randomly selected 11 from 31 provinces as the treatment group, and the others as the control group, constructed a new DID variable for the placebo test, and repeated this process 500 times and 800 times, respectively. The results of the placebo test are shown in Figure 3 and Figure 4. The dotted lines in the figures represent the baseline estimation (−0.012). Two placebo results show that the coefficient of randomization was concentrated near 0, which is significantly different from the baseline estimation. These results show that the negative impact of the snow disaster on GAD does not come from the omitted random variables.

### 5.3. Heterogeneity Analysis

The baseline results show that the snow disaster will reduce the level of GAD in the long run. GAD levels have fallen by an average of 3.07% in the 10 years since the snow disaster. Due to China’s vast territory, there are significant differences in climate and economic factors in different regions, which will cause differences in the impact of snow disaster on GAD. Therefore, we analyzed the heterogeneity from the four aspects of climate, poverty, industrial structure, and economic development differences.

#### 5.3.1. Climate Zone

Due to the impact of climate shocks, there is heterogeneity in different precipitation areas [13,17,49] and, according to the climate characteristics of China, we divided the samples into humid areas (annual precipitation > 800 mm), semi-humid areas (annual precipitation 400–800 mm), and semi-arid areas (annual precipitation < 400 mm). The results are shown in Table 7, columns (1)–(3).

Columns (1)–(3) in Table 7, respectively, report the results of humid, semi-humid, and semi-arid areas. The results showed that after the snow disaster, the GAD index of the humid area and the semi-humid area dropped by 0.008 and 0.017, respectively, while the GAD index of the semi-arid area increased by 0.008. Combined with the average value of the sample, the snow disaster reduced the GAD level in the humid area by 2.12%, the GAD level in the semi-humid area by 4.29%, and the GAD level in the semi-arid area by 2%. This result indicates that the impact of snow disasters on GAD has significant heterogeneity in different climate regions. The inhibitory effect of snow disasters on GAD mainly occurs in areas with abundant rainfall. The reason is that these areas are prone to floods due to fluctuations in precipitation, causing greater damage to agriculture.

#### 5.3.2. Poor and Non-Poor Counties

Agricultural development is greatly affected by government policies. In China, poor and non-poor counties will receive varying degrees of government subsidies. Therefore, the impact of the snow disaster on the GAD may be heterogeneous due to differences in government subsidies and policies. According to the list of poor counties announced by the Chinese government, we divided the sample into poor counties and non-poor counties for heterogeneity analysis. The results are shown in Table 7, columns (4)–(5). The results show that the snow disaster reduced the GAD index by 0.005 and 0.018 for poor and non-poor counties, respectively, and their GAD levels by 1.28% and 4.63%, respectively. Therefore, there is no significant heterogeneity. However, because the absolute value of the coefficient in poor counties is significantly lower than in column (5), this indicates that the impact of the snow disaster on the level of GAD is relatively moderate in poor counties.

#### 5.3.3. Agricultural and Non-Agricultural Counties

Counties with different industrial structures may be affected differently when faced with extreme climate shock [42]. To test this difference, we divided agricultural counties and non-agricultural counties for analysis according to the proportion of agricultural employment in each county. The results are shown in Table 8, columns (1)–(2).

The results show that in both agricultural and non-agricultural counties, the impact of the snow disaster on GAD is significantly negative, with a coefficient of −0.01 for agricultural counties and −0.014 for non-agricultural counties, and the impact is similar. These results show that the impact of extreme climate shocks on GAD does not have significant heterogeneity in agricultural and non-agricultural counties. The reason for this is that the extent of the snow disaster is not influenced by the industrial structure and therefore there is no heterogeneity in the impact of the snow disaster.

#### 5.3.4. Economic Development

Finally, considering the different levels of development in different economic regions, we divided the sample into eastern, central, and western regions for heterogeneity analysis. The eastern part has the highest level of economic development and the western part has the lowest level [37]. The results are shown in Table 8, columns (3)–(5). The results show that in the eastern and middle regions, the coefficients of $Rain_{i}^{s}\times Post_{t}$ are −0.031 and −0.015, respectively, which are both significant at the 1% level. It shows that the impact of the snow disaster on the level of GAD is similar in the two regions. In the western region, the coefficient of $Rain_{i}^{s}\times Post_{t}$ is not significant and close to 0, indicating that the snow disaster had almost no impact on GAD.

The above results indicate that the impact of the snow disaster on GAD has significant heterogeneity in regions with different economic development levels. The snow disaster reduced the GAD index by 0.031 and 0.015 in the eastern and central regions, respectively, and had no effect on the GAD index in the western region. The reason is that farmers are more bound by land, and climate shocks affect agricultural output more than the green level in the western region.

## 6. Mechanism Analysis

The baseline results show that the snow disaster has significantly reduced the level of GAD. According to the conclusions of the theoretical analysis part, extreme climate shocks lead to lower agricultural incomes, causing agricultural production to be exposed to long-term climate risks, changing farmers’ production decisions. To verify this mechanism, we conducted empirical tests from three aspects of farming willingness, production energy consumption, and agricultural pollution. The results are shown in Table 9.

### 6.1. Farming Willingness

Extreme climate shocks have led to a decline in agricultural output and damage to agricultural infrastructure, resulting in a widening GAD between agricultural income and non-agricultural income. The opportunity cost of engaging in agricultural production has increased. Therefore, the snow disaster will reduce the farming willingness of farmers in the disaster-stricken areas, causing them to transfer or abandon farmland, thereby reducing the utilization rate of cultivated land. The reduction in the utilization rate of arable land has reduced the degree of farmland improvement, resulting in soil erosion, and may lead to the occupation of part of the arable land, which is not conducive to the GAD. Based on the above analysis, we use the utilization rate of cultivated land as a mechanism variable to test this effect, and the results are shown in Table 9, column (1).

The results show that the coefficient of $Rain_{i}^{s}\times Post_{t}$ is −2.013 and is significant at the level of 1%, which indicates that the snow disaster has significantly reduced the utilization rate of cultivated land. Specifically, after the snow disaster, the utilization rate of cultivated land dropped by an average of 2.01%. Due to the irreplaceability of land in agricultural production, the decline in the utilization rate of arable land will decrease the total input of agricultural production, thereby reducing the total agricultural output. At the same time, the deserted land leads to barrenness of part of the farmland, causing soil erosion and desertification, which is not conducive to environmental protection and GAD.

### 6.2. Production Energy Consumption

Based on theoretical analysis, farmers who continue to engage in agricultural production will be more inclined to pursue short-term gains due to greater uncertainty. One way to pursue short-term gains in agriculture is to increase production input, such as water resources, electricity, and other consumable resources, which leads to an increase in the energy consumption of agricultural production. Based on this, we use the variables of diesel, electricity, and water resource input per unit of agricultural output, and use the entropy method to construct the agricultural production energy consumption index as a mechanism variable for testing. The results are shown in Table 9, column (2).

The results show that the coefficient of $Rain_{i}^{s}\times Post_{t}$ is 0.003 and is significant at the level of 1%. The snow disaster has increased the energy consumption index of agricultural production by 0.003. Considering that the average energy consumption of agricultural production is 0.037, the snow disaster has increased the energy consumption of agricultural production by 8%. This result shows that after extreme climate shocks, farmers are more inclined to pursue short-term gains due to greater climate risks in the future, and will increase agricultural water input. At this time, excessive extraction of river water and groundwater became the choice of many farmers. The diesel and electricity required to start the pumps have led to an increase in resource consumption, which ultimately leads to an increase in the total energy consumption of agricultural production. These actions will have a negative environmental impact on economic development and reduce the level of GAD.

### 6.3. Agricultural Pollution

Based on theoretical analysis, the snow disaster will increase the long-term climate risk and decrease farmers’ willingness to protect the environment. This risk causes farmers to invest more fertilizers and pesticides to ensure short-term benefits. We use the logarithm of the input of fertilizer and pesticide per hectare as the mechanism variable for analysis to test this mechanism. The results are shown in Table 9, columns (3)–(4).

The results show that the snow disaster increased the fertilizer input per hectare by 1.3% and the pesticide input by 4.7% on average, which led to increased pollution in agricultural production and lowered the level of GAD. Increased inputs of fertilizers and pesticides indicate that farmers are more inclined to guarantee short-term harvests rather than pursue long-term gains when facing greater climate risks in the future. This result further validates the conclusion that extreme climate shocks affect the GAD by changing farmers’ risk expectations.

As farmers are more inclined to guarantee short-term harvests after extreme weather shocks, extreme weather may not reduce the total output of agricultural products. However, due to the long-term decline in the level of GAD, the reduction in green agricultural products will reduce the overall quality of agricultural products. To verify this conjecture, we use the total output of major agricultural products and primary industry value-added as the explained variables to study the impact of extreme climate shocks on the output and quality of agricultural products. The relevant results are shown in Table 10.

Columns (1)–(3) in Table 10, respectively, take the output of main agricultural products, the per capita output of agricultural products, and the primary industry value-added (all logarithms) as the dependent variables. The results show that after the snowstorm, the output of major agricultural products increased by 6.8%, but the value of agricultural output dropped by 1.9%. This result shows that the value of agricultural products per unit has fallen by 8.15%. Although the output of agricultural products increased after the snow disaster, the quality of agricultural products declined, which led to a decrease in their value. This result verified our conjecture. After the snow disaster, farmers pursue short-term harvests and increase the input of fertilizers and pesticides to increase the output of agricultural products. However, due to the reduction in the proportion of green agricultural products, the overall quality of agricultural products in the long-term declines and the agricultural output value decreases.

## 7. Conclusions and Policy Implications

Extreme climate shocks cause agricultural yield reductions and increase long-term climate risk, altering farmers’ long-term production decisions and affecting GAD. This impact is more significant with the increased levels of climate change. We use the difference-in-difference method to study the impact of extreme climate shocks on GAD based on the context of the 2008 snow disaster in China.

The conclusion shows that extreme climate shocks are one of the most important factors hindering GAD. Additionally, the impact of extreme climate shocks is heterogeneous across climate and economic zones. Taking targeted measures to cope with extreme climate shocks is an important guarantee to promote GAD in China against the background of intensifying climate change and more frequent extreme weather. Meanwhile, we find that the mechanism by which extreme climate shocks reduce GAD is mainly to change farmers’ income and expectations and extreme climate shocks have reduced the quality of agricultural products in the long term. Therefore, government subsidies and agricultural insurance can mitigate this effect and improve the quality of GAD.

Based on the above findings, we propose the following policy recommendations: (1) Increasing investment in agricultural infrastructure to enhance farmers’ capacity to cope with climate disasters. (2) Adapting to local conditions and making appropriate agricultural policies in different regions, focusing on post-disaster reconstruction and security in areas with plentiful rainfall and developed regions. (3) Providing agricultural disaster insurance to protect against future extreme weather disasters, reducing farmers’ expected risks, and protecting the income of green agriculture in the long term. (4) Subsidizing farmers who adopt green production and increase the purchase price of green agricultural products. Subsidies can compensate for the losses caused by extreme climate shocks. Increasing the purchase price of green agricultural products can encourage farmers to continue to adopt green production methods, thus safeguarding the quality of agricultural products and promoting GAD.

## Figures and Tables

**Figure 1 ijerph-18-12055-f001:**
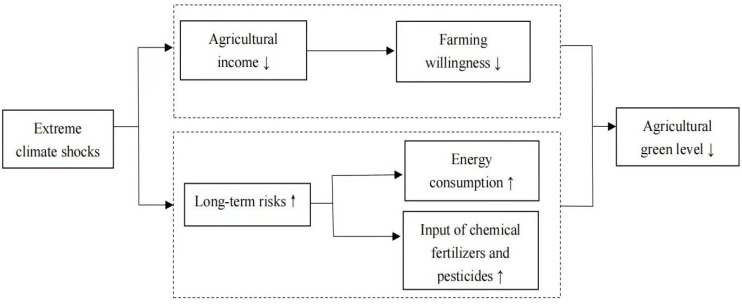
Impact of extreme climate shocks on green agricultural development (GAD).

**Figure 2 ijerph-18-12055-f002:**
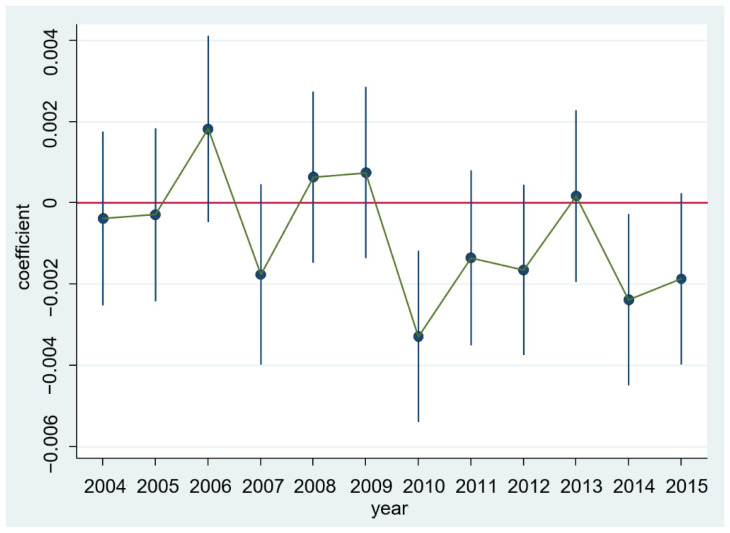
Parallel trend test of GAD index.

**Figure 3 ijerph-18-12055-f003:**
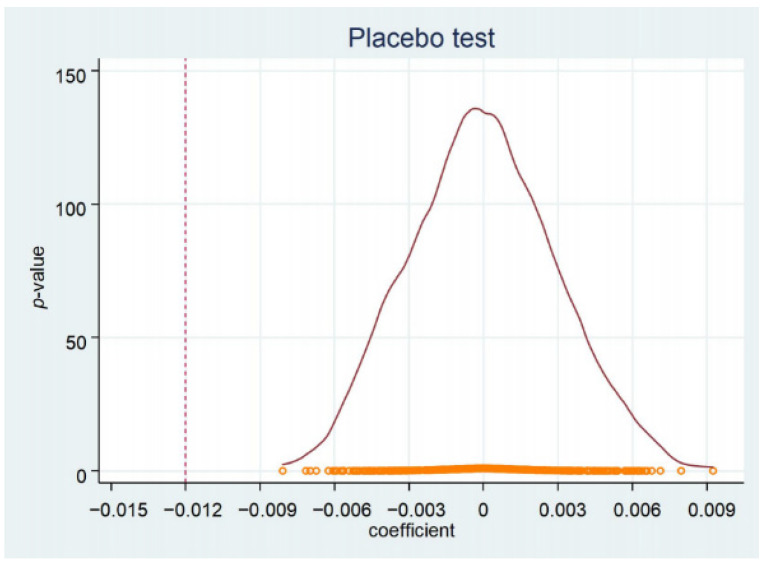
Placebo test (500 times). Notes: The red dot line represents the distribution function of the estimated coefficient. The same below.

**Figure 4 ijerph-18-12055-f004:**
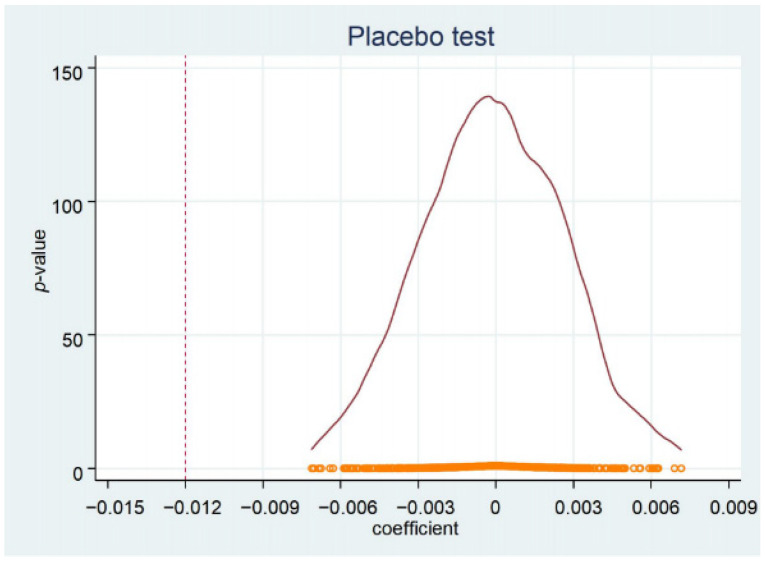
Placebo test (800 times).

**Table 1 ijerph-18-12055-t001:** Calculation of green agricultural development (GAD) index.

Primary Indicators	Secondary Indicators	Measurement Method	Direction
Agricultural endowment	Arable land	Total arable land area	Positive
	Water resources	Water resources per capita	Positive
	Forest resources	Forest coverage rate	Positive
Agricultural production efficiency	Agricultural productivity	Per capita output of major agricultural products	Positive
	Agricultural output efficiency	Primary industry value-added/Agricultural population	Positive
	Arable land utilization	Sown area/arable land area	Positive
	Effective irrigation rate	Effective irrigated area/Sown area	Positive
	Mechanization level	Agricultural machinery power/Sown area	Positive
Agricultural energy consumption	Diesel consumption	Agricultural diesel consumption/Primary industry value-added	Negative
	Electricity consumption	Electricity for agriculture/Primary industry value-added	Negative
	Water consumption	Water for agriculture/Primary industry value-added	Negative
Agricultural pollution	Fertilizer input	Total fertilizer input/Sown area	Negative
	Pesticide input	Total pesticide input/Sown area	Negative
	Plastic film input	Total agricultural plastic film input/Sown area	Negative
Environmental protection	Afforestation area	Afforestation area	Positive
	Erosion control	Soil erosion control area	Positive
	Farmland governance	Flood removal area/Sown area	Positive

**Table 2 ijerph-18-12055-t002:** Main variable definitions and data sources.

Variable	Definition of Variables	Data Sources
Green	Green agricultural development index	China County Statistical Yearbook, China Statistical Yearbook & Statistical Yearbook of every province
Rain	Rainfall fluctuations	FLDAS Noah Land Surface Model L4 Global Monthly
Production	Output per capita of major corps	China County Statistical Yearbook and China Rural Statistical Yearbook
Perfirst	Per capita primary industry value-added	China County Statistical Yearbook
Arable rate	Arable land utilization rate	China Rural Statistical Yearbook and Statistical Yearbook of every province
Fertile	Fertilizer input per hectare	China Rural Statistical Yearbook and Statistical Yearbook of every province
Pesticide	Pesticide inputs per hectare	China Rural Statistical Yearbook and Statistical Yearbook of every province
POP	Total population	China County Statistical Yearbook
Machinery	Total agricultural machinery power	China County Statistical Yearbook
First	Primary industry value-added	China County Statistical Yearbook
Second	Secondary industry value-added	China County Statistical Yearbook
Expend	Total government expenditure	China County Statistical Yearbook
Invest	Total social investment	China County Statistical Yearbook

Note: Abbreviation POP denotes population. FLDAS is Famine Early Warning Systems Network (FEWS NET) Land Data Assimilation System.

**Table 3 ijerph-18-12055-t003:** Descriptive statistics for the main variables.

Variable	Affected Counties		Non-Affected Counties	
	Mean	SD	Mean	SD
Green	0.39	0.18	0.39	0.17
Production	509	284	642	531
Perfirst	3156	2689	3847	3847
Arable rate	146	51	145	52
Fertile	347	133	312	103
Pesticide	10.28	7.90	9.99	6.68
POP	478	386	476	325
Machinery	356	4362	430	5145
First	1475	1594	1696	1617
Second	4903	10,867	5662	9868
Expend	1510	1967	1576	1971
Invest	6245	10,720	7067	11,532

Note: The meaning, calculation method, and data source of each variable are shown in Table 2. Green is an index and has no unit. The units of other variables are as follows: Production (kg per capita); Perfirst (Renminbi (RMB) per capita); Arable rate (%); Fertile and Pesticide (kg per hectare); POP (thousands of people); Machinery (million watts); First, Second, Expand, and Invest (million RMB).

**Table 4 ijerph-18-12055-t004:** Baseline estimations: Impact of the snow disaster on the GAD index.

Variables	(1)	(2)	(3)	(4)	(5)	(6)
	Green	Green	Green	Green	Green	Green
$Rain_{i}^{s}\times Post_{t}$	−0.008 ***	−0.013 ***	−0.013 ***	−0.012 ***	−0.012 ***	−0.018 ***
	(0.001)	(0.001)	(0.001)	(0.001)	(0.001)	(0.002)
POP			−0.007	0.012	0.012	0.004
			(0.008)	(0.008)	(0.009)	(0.009)
Machinery			0.003	0.007 ***	0.007 ***	0.007 ***
			(0.002)	(0.002)	(0.002)	(0.002)
First			−0.031 ***	−0.028 ***	−0.029 ***	−0.027 ***
			(0.002)	(0.002)	(0.002)	(0.002)
Second			−0.005 ***	0.001	0.001	0.001
			(0.001)	(0.001)	(0.001)	(0.002)
Expend				−0.042 ***	−0.043 ***	−0.043 ***
				(0.002)	(0.002)	(0.002)
Invest					0.001	0.0004
					(0.001)	(0.001)
County fix	No	Yes	Yes	Yes	Yes	Yes
Time fix	Yes	Yes	Yes	Yes	Yes	Yes
Obs	39,270	39,270	38,354	38,331	38,142	38,186
R^2^	0.694	0.695	0.698	0.700	0.701	0.699
Counties	2078	2078	2077	2077	2077	2086

Notes: *** denotes significance at 1%. All control variables, individual fixed effects, and time fixed effects are included in all specifications. Obs denotes observations.

**Table 5 ijerph-18-12055-t005:** Considering time trend, municipalities, concurrent events, and climatic factors.

Variables	(1)	(2)	(3)	(4)	(5)
	Time Trend	Municipalities	Land Transfer	Rainfall1	Rainfall2
	Green	Green	Green	Green	Green
$Rain_{i}^{s}\times Post_{t}$	−0.012 ***	−0.012 ***	−0.011 ***	−0.012 ***	−0.013 ***
	(0.001)	(0.001)	(0.001)	(0.001)	(0.001)
Obs	38,142	37,440	32,140	38,006	38,006
R^2^	0.701	0.702	0.413	0.707	0.705
Counties	2077	2040	2076	2077	2077

Notes: *** denotes significance at 1%. All control variables, individual fixed effects, and time fixed effects are included in all specifications. Obs denotes observations.

**Table 6 ijerph-18-12055-t006:** Considering different indicators and methods.

Variables	(1)	(2)	(3)	(4)	(5)
	PSM-DID	Identify2	2SLS-Rainfall	2SLS-lat&lon	Index2
	Green	Green	Green	Green	Input
$Rain_{i}^{s}\times Post_{t}$	−0.013 ***	−0.024 ***	−0.034 ***	−0.041 ***	−0.006 ***
	(0.001)	(0.002)	(0.004)	(0.002)	(0.0004)
Obs	34,500	38,186	38,053	38,112	38,142
R^2^	0.726	0.700	0.701	0.703	0.800
Counties	2077	2086	2067	2075	2077

Notes: Econometric methods: PSM-DID (propensity score matching difference in difference method); 2SLS (two stage least square method). Abbreviation lat&lon denotes longitude and latitude, see the text for specific explanations. *** denotes significance at 1%. All control variables, individual fixed effects, and time fixed effects are included in all specifications. Obs denotes observations.

**Table 7 ijerph-18-12055-t007:** Heterogeneity analysis 1: Climate and poverty.

Variables	Climate Zone			Poor/Non-Poor	
	(1)	(2)	(3)	(4)	(5)
	Humid	Semi-Humid	Semi-Arid	Poor	Non-Poor
	Green	Green	Green	Green	Green
$Rain_{i}^{s}\times Post_{t}$	−0.008 ***	−0.017 ***	0.008 ***	−0.005 ***	−0.018 ***
	(0.001)	(0.002)	(0.003)	(0.002)	(0.001)
Obs	18,858	14,373	4775	13,891	24,251
R^2^	0.751	0.718	0.783	0.684	0.714
Counties	1028	767	282	761	1316

Notes: *** denotes significance at 1%. All control variables, individual fixed effects, and time fixed effects are included in all specifications. Obs denotes observations.

**Table 8 ijerph-18-12055-t008:** Heterogeneity analysis 2: Industrial structures and economic development.

Variables	Agriculture/Non-Agriculture		East/Middle/West		
	(1)	(2)	(3)	(4)	(5)
	Agriculture	Non-Agriculture	East	Middle	West
	Green	Green	Green	Green	Green
$Rain_{i}^{s}\times Post_{t}$	−0.010 ***	−0.014 ***	−0.031 ***	−0.015 ***	0.001
	(0.001)	(0.002)	(0.002)	(0.002)	(0.001)
Obs	20,176	17,877	10,541	11,419	16,182
R^2^	0.690	0.718	0.719	0.726	0.736
Counties	1108	959	563	611	903

Notes: *** denotes significance at 1%. All control variables, individual fixed effects, and time fixed effects are included in all specifications. Obs denotes observations.

**Table 9 ijerph-18-12055-t009:** Mechanism analysis of the snow disaster on GAD.

Variables	(1)	(2)	(3)	(4)
	Arable Rate	Consumption	Fertilizer	Pesticide
	Arable Rate	Consumption	lFertile	lPesticide
$Rain_{i}^{s}\times Post_{t}$	−2.013 ***	0.003 ***	0.013 ***	0.047 ***
	(0.266)	(0.0003)	(0.001)	(0.002)
Obs	37,375	38,142	38,142	38,142
R^2^	0.255	0.638	0.717	0.493
Counties	2077	2077	2077	2077

Notes: lFertile and lPesticide denote the logarithm of the input of fertilizer and pesticide per hectare. *** denotes significance at 1%. All control variables, individual fixed effects, and time fixed effects are included in all specifications. Obs denotes observations.

**Table 10 ijerph-18-12055-t010:** The impact of the snow disaster on the output and quality of agricultural products.

Variables	(1)	(2)	(3)
	Products	Per Products	Primary Value
	lProduction	lpro	lFirst
$Rain_{i}^{s}\times Post_{t}$	0.068 ***	0.068 ***	−0.019 ***
	(0.004)	(0.004)	(0.002)
Obs	38,140	38,140	38,142
R^2^	0.074	0.093	0.875
Counties	2077	2077	2077

Notes: lProduction, lpro and lFirst denote the output of main agricultural products, the per capita output of agricultural products, and the primary industry value-added (all logarithms). *** denotes significance at 1%. All control variables, individual fixed effects, and time fixed effects are included in all specifications. Obs denotes observations.

## Data Availability

The data presented in this study are available on request from the corresponding author. The data are not publicly available due to the privacy of Chinese counties.

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
