# Peer review of "Extreme Climate Shocks and Green Agricultural Development: Evidence from the 2008 Snow Disaster in China"

_ijerph, 2021, doi:10.3390/ijerph182212055_

Round 1

Reviewer 1 Report

The paper reports on the effect of extreme climate shock's on green agricultural development.

The paper is in genreal well written and interesting, however some improvements might be considered before publication. 

The author introducte a numebr os statistics, summarizede in Table 3. The meaning and the uncertainty of specific variables should be better discussed. Additionally, it is not clear how they have specifically computed, and what is the meaningfulness of values reported even with 8 or 9 meaningful digits. Meaning should be described with more detail and number approximation should be revised.

Many variables seem quite weak. By way of example "Pesticide inputs per hectare" "machinery power" or "Fertilizer input per hectare" need further description to be considered. Organic fertilizer and mineral feertilizer have totally different efficiency; pesticide effectiveness depends on the distribution method and timeliness; machinery power should be considered along with the age of implemnts or tractors. 

Geographical characteristics should be considered (the areea is hilly, so thee slope plays a critical role). 

A critical role is played by the amount of available role and by irrigation practices, which aree only marginally mentioned. 

Also education is important in agricultural development, but has not been considered in the paper.

In general all of the variable need some deeper analysis, otehrwise the comments can be only very generic. 

The graphical part must be improved, provding also a map of the interested area, and also of the test areas included in the "Placebo test".

References should be improved. see e.g.:

Socio-economic impact of and adaptation to extreme heat and cold of farmers in the food bowl of Nepal    Budhathoki, N.K., Zander, K.K.    2019    International Journal of Environmental Research and Public Health
Factors associated with seasonal food insecurity among small-scale subsistence farming households in rural honduras    Dodd, W., Cerna, M.G., Orellena, P., (...), Kipp, A., Cole, D.C.    2020    International Journal of Environmental Research and Public Health
Climate change impacts on agricultural production and crop disaster area in China    Shi, Z., Huang, H., Wu, Y., Chiu, Y.-H., Qin, S.    2020    International Journal of Environmental Research and Public Health
Heat-moderating effects of bus stop shelters and tree shade on public transport ridership    Lanza, K., Durand, C.P.    2021    International Journal of Environmental Research and Public Health
Extreme weather events in agriculture: A systematic review    Cogato, A., Meggio, F., Migliorati, M.D.A., Marinello, F.    2019    Sustainability (Switzerland)

Author Response

First of all, please allow us to take this opportunity to express our heartfelt thanks to you for taking time out from your busy schedule to review this manuscript. You have provided us with constructive comments and suggestions, which are of great help for us to further improve this manuscript. We have carefully reviewed and revised the manuscript according to your valuable comments and suggestions. Here, we explain the revised work in detail below and provide the point-by-point responses to the reviewers' comments.

Reviewer 2 Report

All my comments, questions and suggestions to improve the manuscript are in sticky notes in the relevant parts in the pdf file attached.

Author Response

(The authors gave the same response as above.)

Round 2

Reviewer 1 Report

Authors have properly improved the paper. 

I just still believe the number of meaningful digits should be reduced: e.g. it is nosense to write "mean= 3155.59 st. dev.=2688.86". Since the standard deviation is somehow a measure of the unceertainty I would write just 3160 and 2690 or 3200 and 2700. In general I would recommend to revise the number of meaningful digits according to their actual significance and robustness. 

Author Response

Thanks again for taking time out from your busy schedule to review this manuscript. You have provided us with constructive comments and suggestions, which are of great help for us to further improve this manuscript. We have carefully reviewed and revised the manuscript according to your valuable comments and suggestions. Here, we explain the revised work in detail below and provide the point-by-point responses to the reviewer's comments. Please see the attachment.

Reviewer 2 Report

The authors significantly improved the manuscript taking all my suggestions and corrections into consideration.

Author Response

Thanks for your approval of our revision work. Please allow us to take this opportunity to express our heartfelt thanks to you again. You have provided us with constructive comments and suggestions, which are of great help for us to further improve this manuscript.